# EAT: EXPERT ACCOUNT TRACKER FOR EFFICIENT MOE INFERENCE

## ABSTRACT

Mixture-of-Experts (MoE) models have emerged as a revolutionary method to scale Transformer models. However, traditional MoE architecture still suffers from inefficiency since a large number of experts are unnecessarily activated. Existing approaches for reducing the number of activated experts often overlook the historical performance of each expert. In this paper, we propose EAT, a novel method called **Expert Account Tracker (EAT)**, which utilizes history-awareness metrics and adaptive thresholding to dynamically select the most important experts, thereby reducing the activated expert number while effectively maintaining the model performance. Experiments show that EAT outperforms the existing baseline Top-P method across multiple models and datasets, achieving over 25% an average reduction compared to the vanilla method in the number of activated experts and performing better token generation speed compared to the baseline. Additionally, through ablation studies, we find that excessively reducing the number of activated experts can significantly harm model performance, and the importance of experts varies across layers, with higher-level experts being generally more critical.

## 1 INTRODUCTION

Mixture-of-Experts (MoE) architecture has emerged as a promising solution to efficiently enlarge both Large Language Models (LLMs). This architecture replaces the standard Feed-Forward Network (FFN) in each Transformer block with a set of specialized "expert" networks and then leverages a trainable router to selectively activate the most relevant experts for each input token. Through sparse activations, MoE architecture can improve model scalability without much increase in the inference time compared to dense models. This allows the deployment of much larger models in real-world applications while maintaining acceptable computational overhead.

Despite the inherent sparsity advantages of MoE, current activation strategies still suffer from the following limitations: (1) Lack of flexibility in the vanilla strategy. Such strategies select a fixed number of experts, regardless of their individual importance and the specific features of the input. This often leads to redundant activation, compromising the model's ability to achieve an optimal balance between computational efficiency and performance; (2) Neglect of historical expert performance. Some dynamic expert selection methods (e.g., Top-P activation) fail to consider the historical performance of experts. This makes it difficult to differentiate between "consistently reliable experts" and "noisy experts". These limitations hinder the achievement of increased sparsity and can lead to performance degradation when attempting to reduce the number of activated experts.

To address these issues, we propose a novel strategy called **E**xpert **A**ccount **T**racker (**EAT**), which employs a history-aware approach with adaptive thresholding. EAT first establishes an importance score for each expert based on its historical performance. This score is a comprehensive metric that considers multiple aspects, such as the frequency of expert activation, the total gate weights it has been assigned, and its contribution to the model's final score. This approach allows EAT to evaluate each expert's long-term value more accurately, rather than just its immediate score for a given input. Additionally, the score of its contribution to the model is not a fixed value but is smoothly updated to ensure that the experts' contribution is consistently evaluated, balancing its immediate output with its long-term performance history, which in turn provides a stable and reliable foundation for the overall importance score. During inference, we define the final importance score for each expert by

taking a weighted sum of the router-assigned gating score and its history-based importance score. Our dynamic thresholding strategy then employs a dual-constraint strategy, first choosing a set of candidate experts and then applying a Top-K constraint to select the final subset of experts to be activated, thereby adaptively pruning redundant experts and maintaining the model performance.

To validate the effectiveness of our proposed EAT method, we test it on popular MoE models, and EAT outperforms the baseline Top-P models across various benchmarks under different modes while achieving a 25% reduction in the number of activated experts compared to the vanilla models. The token generation speed of EAT models also exceeds the baseline Top-P models, which demonstrates the perfect balance of inference efficiency and model performance of our method. Our key contributions are summarized as:

- We introduce Expert Account Tracker (EAT), an efficient MoE activation strategy that combines a historical awareness mechanism with an adaptive thresholding approach. It dynamically selects experts based on a final importance score, which is a combination of their historical performance and current routing probability.
- Through extensive systemic experiments on MoE models, the EAT method demonstrates superior performance compared to Top-P baseline models across various benchmarks. It achieves a 25% reduction in the number of activated experts while also exceeding the token generation speed of the baselines.
- We conduct in-depth ablation studies that evaluate the relationship between model PPL, the number of activated experts, and different model layers. This provides valuable inspiration and insights for future research about optimizing expert activation strategies.

## 2 RELATED WORKS

Existing acceleration methods for well-trained MoE LLMs can be divided into two main categories: Parameter Compression and Acceleration In The Inference Stage.

### 2.1 PARAMETER COMPRESSION

The objective of this category is to reduce the overall number of parameters in MoE LLMs. This can be understood as a static optimization strategy that permanently modifies the model's structure (Lu et al., 2024), a core difference from our dynamic method proposed in this paper. Common methods include Merging Experts and Dropping Experts. Moe-pruner (Xie et al., 2024) prunes weights based on router weight information and weight magnitudes. Channel Merging (Zhang et al., 2025) reduces the number of experts by merging their parameters based on similarity. Sub-MoE (Li et al., 2025) prunes experts that have particularly low usage. MoE-I$^2$ (Yang et al., 2024a) uses a two-stage MoE compression framework to reduce model size and computational cost. While these strategies are effective at lowering the scale of models, this compression often disrupts the original expert diversity and arrangement, which causes a significant drop in model performance. As a result, large-scale pre-training or distillation is often necessary to restore performance, leading to considerable computational cost to the overall process.

### 2.2 ACCELERATION IN THE INFERENCE STAGE

This approach aims to reduce computation during the inference stage through more sparse activation, without altering the model's parameters. CMoE (Pei et al., 2025) reorganizes all dense FFN parameters into a combination of "shared experts" and "routed experts". Dynamic MoE (Huang et al., 2024) sets a threshold based on routing probabilities, only keeping experts with higher probabilities. XMoE (Yang et al., 2024b) enables each token to autonomously determine the number of experts to activate by using a predefined probability threshold. Ada-K (Yue et al., 2024) uses a learnable allocator module and a reinforcement learning framework algorithm to adjust the number of activated experts. Expert Pruning And Skipping (Lu et al., 2024) skips less important experts when the routing weight of the second-best expert is less than a specific ratio of the routing weight of the best expert. However, a key limitation of these methods is their reliance on the routing distribution of the current input, which overlooks the historical performance of experts and can lead to an unstable selection process.

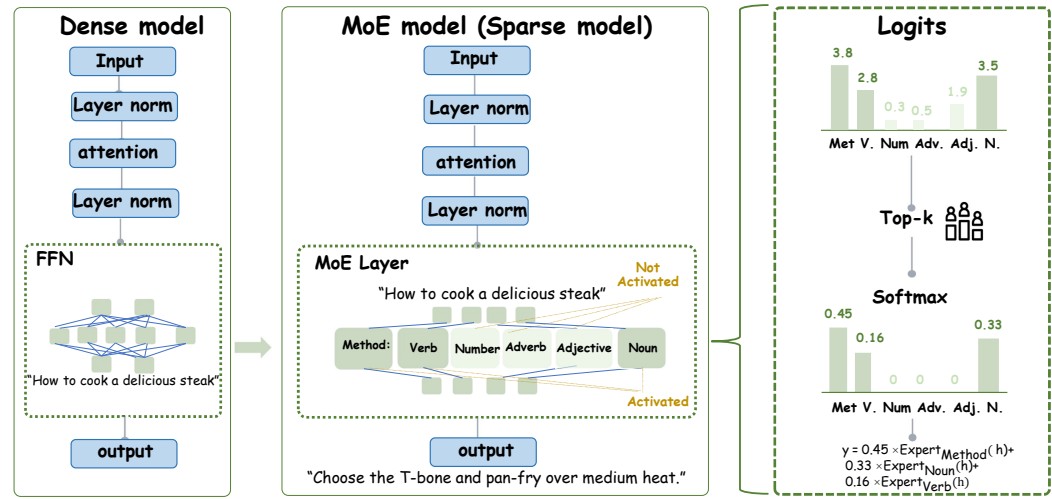

Figure 1: Overview of the dense model and MoE model. As a sparse model, the MoE architecture distributes tokens to specific experts for processing. It is different from the traditional dense models, which activate and utilize all parameters.

This paper focuses on **Acceleration In The Inference Stage**. Unlike existing methods, our proposed EAT is a novel sparse MoE activation strategy that combines experts' historical performance with current routing probabilities and uses dynamic and adaptive thresholding to select experts. This method aims to solve the inefficiency issue of traditional approaches and select the most appropriate experts for the tasks, maintaining the model performance while significantly reducing redundant experts and greatly improving inference efficiency without extra retraining costs.

## 3 METHOD

### 3.1 PRELIMINARY

To provide a clear understanding of our proposed method, we first introduce the fundamental architecture of MoE models. MoE LLM is a neural network architecture designed to enhance model efficiency and performance through a "divide-and-conquer" strategy. As depicted in the architectural overview of MoE in Figure 1, the MoE architecture differs fundamentally from traditional dense models by adopting a sparse activation mechanism. It processes complex tasks by routing inputs to specialized sub-networks (i.e., "experts"), and dynamically schedules resources via a "gating network". Compared with traditional dense models, it utilizes sparse activation: each expert (typically FFN) focuses on processing specific types of inputs (such as syntactic features, semantic information, and domain-specific knowledge) based on input features.

Structurally, MoE models primarily consist of two key components: the Expert Module and the Gating Network. (1) Expert Module is a set of independent sub-networks that replaces the standard FFN layers in the traditional Transformer (Vaswani et al., 2017) architecture. Mixtral-8x7B (Jiang et al., 2024) contains 8 experts in total. DeepSeek-V2 (Liu et al., 2024) classifies experts into shared experts and routed experts, which handle general and specialized tasks, respectively. (2) Gating Network, or router, is a small neural network responsible for calculating matching scores between the input tokens and each expert. It then uses a linear layer and a Softmax function to select the Top-K experts for computation (e.g., Top-2, Top-4) and their corresponding weights. The final output is obtained by a weighted fusion of the selected experts' outputs.

**Core Principles and Key Technologies**. After the input tokens pass through the attention layer, the gating network proceeds through three stages: score calculation, Top-K selection, and expert computation with fusion. This process begins with the network applying a linear transformation to the hidden states $h$, to compute a raw score (logit) for each expert: logit = Linear($h$), where the output dimension equals the number of experts. Subsequently, the scores are sorted to retain the

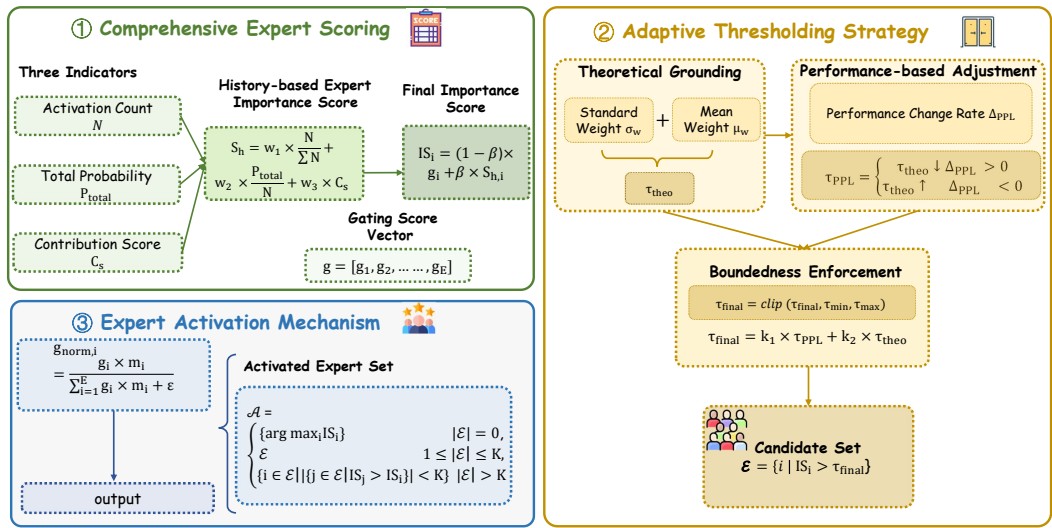

Figure 2: Overview of our proposed EAT architecture. Unlike traditional routing methods that focus solely on the current input, our EAT model is designed to achieve a superior balance between the model performance and the computation cost. EAT selects the most useful experts by continuously tracking their historical contributions and incorporating this information into the routing decision.

Top-K experts, with the remaining scores being set to $-\infty$. Finally, the gate weight $g$ is derived using Softmax: $g = \text{Softmax}(\text{Top-K}(\text{logits}, K))$, which normalizes the scores into a probability distribution. The experts selected then process the tokens and their outputs are weighted and summed according to their corresponding gating weights, then fusing the results into a single output:

$$y = \sum_{i=1}^{K} g_i \cdot \text{Expert}_i, \tag{1}$$

where $y$ represents final output of the MoE layer.

### 3.2 EAT-MOE

In this section, we discuss the structure of EAT-MOE. As illustrated in Figure 2, EAT is a dynamic expert selection strategy that combines comprehensive scoring with an adaptive threshold. Its goal is to achieve a superior balance between inference efficiency and model performance. Unlike the vanilla approach, EAT-MOE dynamically changes the activation of experts based on the comprehensive expert importance, routing weights, and real-time performance feedback. This adaptive approach ensures that only the most relevant and efficient experts are activated, retaining the accuracy of the model and reducing the computation cost.

#### 3.2.1 COMPREHENSIVE EXPERT SCORING

Identifying which experts are of great account to each token is a critical challenge in dynamically adjusting the number of activated experts. To this end, we first propose three indicators to reflect the expert's contribution and efficiency: 1) **Activation count** $N$, defined as the total number of a specific expert (e.g., expert i) being selected for activation, serves as a metric to compare expert utilization frequency 2) **Total probability** $\mathcal{P}_{total}$, which is the sum of the router-assigned weights for the expert across all selections, reflects the cumulative reliance on the expert; 3) **Contribution score** $\mathcal{C}_s$ is an exponentially smoothed estimate of the expert's output magnitude. In closed form, $\mathcal{C}_s = (1 - \alpha) \times \mathcal{C}_s^{prev} + \alpha \times \|o\|$, where $\alpha \in [0, 1]$ is a smoothing coefficient and $\|o\|$ represents the L2 norm of the output of expert. In our experiment, we set $\alpha$ as 0.95. Higher $\mathcal{C}_s$ indicates a stronger expert's contribution to the final output.

Therefore, the history-based expert importance score $\mathcal{S}_h$ is obtained by weighted summation:

$$\mathcal{S}_h = w_1 \times \frac{N}{\sum N} + w_2 \times \frac{P_{total}}{N} + w_3 \times \mathcal{C}_s, \tag{2}$$

where $w_1$, $w_2$, and $w_3$ are three hyperparameters representing weights, which are set to 0.4, 0.4, and 0.2 in our experiments, respectively.

However, an expert with a large history-based score may still have little influence on the final model output when its current router-assigned gating score is small. Therefore, instead of using only the history-based score, we define the final importance score $\mathcal{IS}$ of an expert as the combination of its current gating value and history-based score. Specifically, for each token, the router outputs a gating score vector $\mathbf{g} = [g_1, g_2, ..., g_E]$, where $g_i$ represents the current probability that expert $i$ is selected and $E$ is the total number of experts. Further, we can compute the final importance score of the expert $i$ as follows:

$$\mathcal{IS}_i = (1 - \beta) \times g_i + \beta \times \mathcal{S}_{h,i}, \tag{3}$$

where $\beta$ is the weight coefficient (default is 0.25 in our experiments). This final score balances the current routing pattern and the historical performance of experts.

### 3.2.2 ADAPTIVE THRESHOLDING STRATEGY

This section presents adaptive thresholding ($\tau$) as a substitute for fixed Top-K gating, allowing the model to make mode flexible and cost-effective activation decisions. The threshold selection employs a three-stage strategy: theoretical grounding, performance-based adjustment, and boundedness enforcement.

**Theoretical Grounding.** We introduce an initial threshold $\tau_{\text{theo}}$ based on the statistical distribution of routing weights. This thresholding is designed to achieve a dual objective: it guarantees a minimum level of expert activation while remaining sensitive to the statistical properties of the router's output. It is calculated using the mean weight and standard weight, balancing the average level of activation with the dispersion of the weights: 1) **Mean Weight** $\mu_w$ is the arithmetic mean of routing weights to reflect the overall activation tendency; 2) **Standard Weight** $\sigma_w$ measures the degree of dispersion in weight distribution. The detailed formulas of $\tau_{\text{theo}}$ are:

$$\mu_w = \frac{1}{N} \sum_{i=1}^{N} g_i, \quad \sigma_w = \sqrt{\frac{1}{N} \sum_{i=1}^{N} (g_i - \mu_w)^2}, \tag{4}$$

$$\tau_{\text{theo}} = \max(\alpha_1 \cdot \mu_w, \ \mu_w - \alpha_2 \cdot \sigma_w), \tag{5}$$

where $N$ is the total number of experts, $g_i$ represents the gating score assigned by the router, $\alpha_1$ and $\alpha_2$ are scaling factors for $\mu_w$ and $\sigma_w$, $\alpha_1 \cdot \mu_w$ serves as a safeguard to ensure that the model maintains a minimum level of expert activation even when the router's output is highly dispersed, thereby preventing a sharp decline in performance. By taking the maximum of these two values, our method ensures an adaptive threshold that effectively balances computational efficiency with the preservation of model performance. In the experiment, we set $\alpha_1$ and $\alpha_2$ as 0.8 and 0.5, respectively.

**Performance-based Adjustment.** Since input difficulty and distribution drift over time, a fixed threshold cannot accommodate the non-stationarity of the routing problem. We thus adapt the threshold based on real-time model performance metrics PPL. If the performance changes, the threshold is recalibrated, the PPL change rate $\Delta$PPL is calculated as:

$$\Delta\text{PPL} = \frac{\text{PPL}_{\text{current}} - \text{PPL}_{\text{prev}}}{\text{PPL}_{\text{prev}}}, \tag{6}$$

where $\Delta\text{PPL}_{\text{current}}$ represents the perplexity of the model and $\Delta\text{PPL}_{\text{prev}}$ represents the perplexity at a previous step. If $\Delta$PPL is positive, it indicates that the current perplexity is higher than before (meaning a drop in performance). To activate more experts and improve performance, the threshold is decreased by $\lambda_-$=20%: $\tau_{\text{PPL}} = \tau_{\text{theo}} \cdot (1 + \lambda_+)$; otherwise, it is increased by $\lambda_+$=10% to encourage more sparsity: $\tau_{\text{PPL}} = \tau_{\text{theo}} \cdot (1 - \lambda_-)$.

**Boundedness Enforcement.** To ensure stability and prevent extreme activation or sparsity, the final threshold is constrained to a predefined range, $\tau_{\text{final}} \in [\tau_{\min}, \tau_{\max}]$. In our implementation, $\tau_{\min}$ and $\tau_{\max}$ are set as 10th and 90th percentiles of the routing weight distribution, respectively, ensuring a minimum level of expert activation. Finally, we adopt the exponential smoothing transition to avoid abrupt changes in the threshold caused by performance fluctuations: $\tau_{\text{final}} = \kappa_1 \cdot \tau_{\text{PPL}} + \kappa_2 \cdot \tau_{\text{theo}}$ where $\kappa_1$ and $\kappa_2$ are two hyperparameters representing weights, which are set as 0.7 and 0.3 in our experiments, respectively.

### 3.2.3 EXPERT ACTIVATION MECHANISM

During inference, we use a **dual-constraint activation mechanism** to vary the number of selected experts. Our goal is to maximize efficiency by limiting unnecessary activations while maintaining model performance with a minimum-activation guarantee. To achieve this, we first define a candidate expert set $\mathcal{E}$, comprising experts whose final importance score $\mathcal{IS}$ exceeds the dynamic threshold $\tau_{\text{final}}$: $\mathcal{E} = \{i \mid \mathcal{IS}_i > \tau_{\text{final}}\}$. Then, the activated expert set $\mathcal{A}$ is determined by:

$$\mathcal{A} = \begin{cases} \{\arg\max_i \mathcal{IS}_i\} & |\mathcal{E}| = 0, \\ \mathcal{E} & 1 \leq |\mathcal{E}| \leq K, \\ \{i \in \mathcal{E} \mid |\{j \in \mathcal{E} \mid \mathcal{IS}_j > \mathcal{IS}_i\}| < K\} & |\mathcal{E}| > K, \end{cases} \tag{7}$$

where $K$ is the prescribed maximum number of experts (e.g., Top-2, Top-4). This activation mechanism strikes a balance between computational efficiency and preserving the model capabilities.

Next, we normalize the gating scores of the activated experts to ensure proper fusion:

$$g_{norm,i} = \frac{g_i \cdot m_i}{\sum_{i=1}^{E} g_i \cdot m_i + \epsilon} , \quad m_i = \begin{cases} 1 & \text{Expert}_i \in \mathcal{A}, \\ 0 & \text{otherwise}, \end{cases} \tag{8}$$

where $g_i$ represents the original gating score assigned to the expert $i$ by the router, reflecting the initial, unnormalized importance of that expert for the given input. $m_i$ is a binary mask (0 or 1) that indicates whether the expert $i$ is selected for activation and only experts with a mask score of 1 are included in the normalization and fusion process, and $\epsilon = 10^{-8}$ is a smoothing term added to prevent division by zero.

The final output of the MoE layer is produced through a weighted summation, where each activated expert's output is scaled by its corresponding $g_{norm}$ before summation. This dynamic and weighted fusion mechanism ensures that our EAT-MoE effectively adapts to various input distributions and task requirements, optimizing performance while maintaining computational efficiency.

## 4 EXPERIMENTS

### 4.1 EXPERIMENTAL SETUP

To thoroughly assess the effectiveness of the proposed EAT method, we conduct extensive experiments on popular MoE LLMs and a comprehensive benchmark suite.

**Models** Our experiment utilizes two popular MoE LLMs: **Mixtral-8x7B-v0.1** (Jiang et al., 2024) and **Phi-3.5-MoE-instruct** (Abdin et al., 2024). These models are chosen for their MoE architectures and widespread recognition. By testing EAT on both a large, open-source model like Mixtral-8x7B-v0.1 and a highly performant, smaller-scale instruction-tuned model like Phi-3.5-MoE-instruct, we ensure that our findings on efficiency and performance gains are robust and generalizable across different scales and application types. All experiments are performed on 8 NVIDIA A100 GPUs, providing the necessary computational capacity to handle these large-scale models.

**Benchmarks** To comprehensively evaluate the model performance, we employ the OpenCompass evaluation framework (Contributors, 2023). This framework allows us to categorize our evaluations into five key dimensions: **Reasoning, Language, Knowledge, Examination, and Understanding**. To ensure a broad and representative evaluation, we select specific benchmarks within each category. Reasoning: HellaSwag (HeSw) (Zellers et al., 2019) and PIQA (Bisk et al., 2019). Language: CHID (Zheng et al., 2019) and WSC (Levesque et al., 2012). Knowledge: BoolQ (Clark et al., 2019).

| LLM | Method | Reasoning | | Language | | Know. | Examination | | Und. |
|---|---|---|---|---|---|---|---|---|---|
| | | HeSw | PIQA | CHID | WSC | BoolQ | MMLU | CMMLU | XSum |
| **Mixtral -8x7B -v0.1** | Vanilla | 77.11 | 81.07 | 37.51 | 61.54 | 69.11 | 71.67 | 53.11 | 9.19 |
| | Top-P | 75.72 | 79.71 | 32.85 | 60.58 | 66.15 | 65.38 | 46.74 | 8.70 |
| | EAT | 76.79 | 80.30 | 33.97 | **63.46** | 68.4 | 70.37 | 51.04 | 9.08 |
| **Phi -3.5-MoE -instruct** | Vanilla | 75.17 | 80.2 | 66.75 | 68.27 | 75.32 | 76.64 | 61.03 | 14.68 |
| | Top-P | 68.98 | 78.67 | 60.99 | 71.15 | 69.36 | 74.05 | 55.26 | 11.32 |
| | EAT | 69.00 | 79.87 | 61.00 | 67.31 | **76.42** | 75.00 | 57.46 | 12.46 |

Table 1: The main results of our experiments conducted on the OpenCompass Platform. **Know.** denotes Knowledge and **Und.** denotes Understanding.

| LLM | Method | Reasoning | | Language | | Know. | Examination | | Und. |
|---|---|---|---|---|---|---|---|---|---|
| | | HeSw | PIQA | CHID | WSC | BoolQ | MMLU | CMMLU | XSum |
| **Mixtral -8x7B -v0.1** | Vanilla | 2.00 | 2.00 | 2.00 | 2.00 | 2.00 | 2.00 | 2.00 | 2.00 |
| | Top-P | 1.50 | 1.44 | 1.75 | 1.52 | 1.46 | 1.51 | 1.74 | 1.51 |
| | EAT | 1.47 | 1.46 | **1.44** | 1.50 | 1.45 | 1.47 | 1.48 | 1.47 |
| **Phi -3.5-MoE -instruct** | Vanilla | 2.00 | 2.00 | 2.00 | 2.00 | 2.00 | 2.00 | 2.00 | 2.00 |
| | Top-P | 1.32 | 1.67 | 1.74 | 1.72 | 1.71 | 1.76 | 1.75 | 1.73 |
| | EAT | 1.36 | **1.34** | 1.50 | 1.35 | 1.70 | 1.43 | 1.44 | **1.36** |

Table 2: Comparison of the number of activated experts across different expert activation methods.

| Model | Mixtral-8x7B-v0.1 | | | Phi-3.5-MoE-instruct | | |
|---|---|---|---|---|---|---|
| **Method** | Vanilla | Top-P | EAT | Vanilla | Top-P | EAT |
| **Length = input + max output length** 2048+256 | 0.5214 | 0.5100 | **0.5196** | 1.1319 | 0.9479 | **1.1600** |
| 1024+128 | 0.3931 | 0.3739 | 0.3864 | 0.9295 | 0.9073 | 0.9158 |
| 512+32 | 0.3814 | 0.1596 | 0.3671 | 0.8060 | 0.8719 | **1.1421** |

Table 3: The average generation speed of token. The unit is token/sec.

Examination: MMLU (Hendrycks et al., 2021) and CMMLU (Li et al., 2023). Understanding: XSum (Narayan et al., 2018).

All evaluations were executed using OpenCompass's official scripts, employing two primary evaluation models: **Perplexity (PPL) and Generation (GEN)**. Specifically, we used the GEN mode for CHID, XSum, and WSC, while the PPL mode was applied to BoolQ, HeSw, PIQA, MMLU, and CMMLU. The final accuracy scores for each benchmark are standardized by OpenCompass, with a higher score indicating superior performance.

**Baselines** To provide a meaningful comparison, we select the **Top-P** method (Huang et al., 2024) as our baseline. As both Top-K and Top-P are widely recognized MoE LLM methods, choosing the Top-P method allows us to directly compare our EAT's performance with a method that dynamically adjusts the number of activated experts. To ensure a fair comparison, we set a constant p=0.6 for our Mixtral baseline and p=0.2 for Phi baseline, which ensures the number of activated experts is similar to that of our EAT method.

## 4.2 MAIN RESULTS

The main results are summarized in Table 1, Table 2, and Table 3, which clearly demonstrate that our EAT model effectively balances computational efficiency with model performance across various tasks and models.

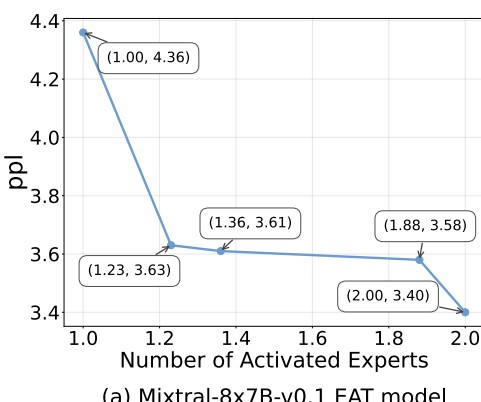 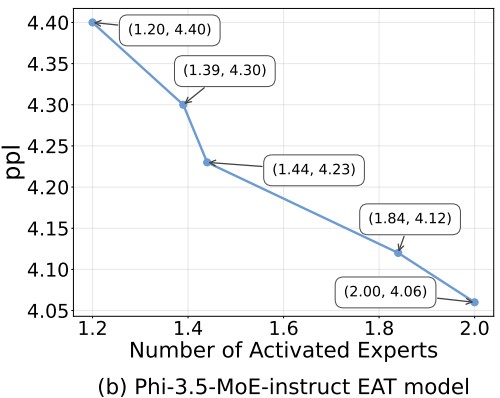

(a) Mixtral-8x7B-v0.1 EAT model  (b) Phi-3.5-MoE-instruct EAT model

Figure 3: Curve of the Relationship Between PPL and Number of Activated Experts.

**EAT Outperforms Baselines in Performance and Efficiency**. As shown in Table 1, the EAT models generally outperform the Top-P baselines across a range of benchmarks and evaluation modes. EAT models consistently achieve higher scores on reasoning, language, and knowledge tasks. Compared to the vanilla model, EAT exhibits a slight decrease in score but activates fewer experts. This is a key finding, as it indicates that the EAT method can achieve performance similar to the vanilla model while being much more computationally efficient. When compared directly to the Top-P method, which activates a similar number of experts, our EAT models show superior performance across every dataset evaluated. This highlights a major advantage of EAT's adaptive approach, which considers an expert's historical performance rather than just relying on its current routing score, ensuring a more effective and higher-quality expert activation.

**EAT Provides a Significant Reduction in Activated Experts**. Table 2 clearly illustrates that EAT models activate far fewer experts than the vanilla models across all categories, achieving a reduction of nearly 25%. On the Mixtral-8x7B-v0.1 model, the vanilla method activates 2 experts for Reasoning tasks, while EAT activates only 1.46 on average. For the Phi-3.5-MoE-instruct model, the number of activated experts for the same Reasoning tasks drops from 2 with the vanilla method to 1.34 with EAT at best. This large reduction in the number of activated experts is a direct result of EAT's dynamic and history-aware selection strategy, which significantly lowers the computational overhead per token.

**Improved Generation Speed**. Beyond the number of activated experts, we also measure the token generation speed, with results presented in Table 3 (token/sec). The EAT model's speed is comparable to the original vanilla models but is faster than the Top-P baselines. For the Mixtral-8x7B model under a 1024 input + 128 max output length setting, the EAT model generates tokens at a rate of 0.3864 tokens/sec, slightly less than the vanilla model (0.3931) but notably faster than the Top-P method (0.3739). For the Phi-3.5-MoE-instruct model in the short sequence setting (512 input + 32 max output length), the EAT model's speed (1.1421 tokens/sec) surpasses both the vanilla model (0.8060 tokens/sec) and the Top-P method (0.8719 tokens/sec). This strongly demonstrates that EAT's optimized inference process provides tangible and large improvements in efficiency.

### 4.3 ABLATION STUDY

Our ablation studies explore two key aspects of the EAT method: the impact of the number of activated experts on model performance and the varying importance of experts across different layers of the network. The models used are the same as the experiments: Mixtral-8x7B-v0.1 and Phi-3.5-MoE-instruct.

**The Impact of the Number of Activated Experts**. We first examine the relationship between the number of activated experts and model performance, measured by **PPL**. As shown in Figure 3, we draw the curve which indicates the relationship between the PPL and the Number of Activated Experts. Notably, as the number of activated experts increases, PPL gradually decreases. Around

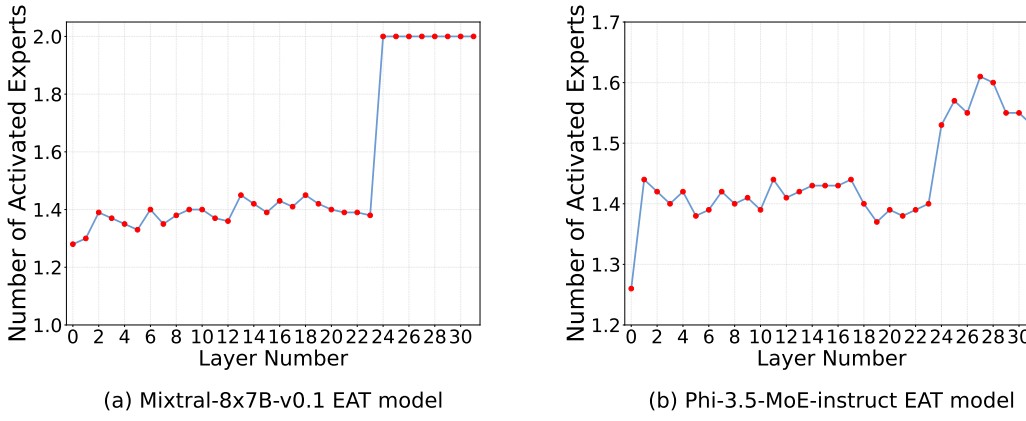

(a) Mixtral-8x7B-v0.1 EAT model      (b) Phi-3.5-MoE-instruct EAT model

Figure 4: Curve of the Relationship Between layer number and Number of Activated Experts.

1.36 [activated experts per token], the PPL reaches a favorable score, and the fluctuation of PPL around this value is relatively small.

A crucial finding is that excessively reducing the number of activated experts can significantly harm model performance. When the number of activated experts is reduced to only one, the PPL for Mixtral-8x7B-v0.1 increases sharply to 4.3, a substantial rise from the more optimal range. This indicates that the drop in activated experts cannot be arbitrary and highlights the importance of maintaining a minimum level of activation to preserve model quality. In Figure 4, we observe that as the number of activated experts increases, the PPL gradually decreases. The PPL curve flattens out around an average of 1.36 activated experts per token, suggesting this is a proper number where the model performance is stable.

**The importance of Experts Across Layers**. Our study also investigates the sparsity and importance of experts across different layers of the MoE models. The results, as presented in Figure 4, reveal a crucial architectural insight: the importance and optimal activation of experts are not uniform across layers.

Specifically, our findings indicate that experts in the lower layers (layers 0 to 23) are less critical compared to those in the higher layers. This observation aligns with the hierarchical nature of Transformer architecture, where lower layers typically learn more general features while higher layers are responsible for handling complex information. Consequently, this feature allows for a more aggressive sparsity strategy. Based on this discovery, we find that we can maintain the number of activated experts in the lower layers within a narrow range (1.3 to 1.5 experts), while allowing for a more liberal expert activation (all or almost Top-K experts) in the higher layers (layers 24 to 30) without significant performance degradation. This insight is valuable for future model design and training, suggesting a potential for more selective and layered sparsity strategies. Instead of applying the vanilla approach, a dynamic, layer-wise routing mechanism can significantly enhance computational efficiency by activating only the necessary number of experts of the model.

## 5 CONCLUSION

In this paper, we introduce EAT, a novel method for optimizing the inference of sparse MoE models. Our approach addresses the limitations of traditional MoE methods, which overlook the historical performance of experts. EAT integrates a history-aware scoring mechanism with a dynamic and adaptive thresholding strategy to select the most appropriate experts for each task. Through extensive experiments across multiple models and datasets, we demonstrate that EAT significantly outperforms existing baselines by activating fewer experts while effectively maintaining the model performance. Our ablation studies provide crucial insights for future model design, revealing that the importance of experts varies across layers and that a reduction in activated experts must be carefully managed to avoid significant performance loss. In summary, EAT represents a robust and flexible strategy that realizes an optimal balance between computational efficiency and model accuracy.

ETHICS STATEMENT

The core of this study focuses on optimizing the inference efficiency of MoE models. EAT method aims to select activated experts by tracking historical expert performance and using an adaptive threshold. We performed all experiments in a controlled, simulated environment using only standard, publicly available datasets. All our experiments were designed to demonstrate that this method can effectively improve efficiency while maintaining model performance. Our study does not involve any potential applications that could raise security or ethical concerns.

REPRODUCIBILITY STATEMENT

To ensure the full reproducibility of our research, we provide our complete experimental code and configuration files available. All other important details can be found in the Experiment section of this paper.

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

# 6 APPENDIX

## 6.1 DECLARATIONS OF LLM USAGE

This paper uses LLMs for polishing and translation of some professional terms, without using LLMs to generate new content.

