# OpenReview forum: "EAT: Expert Account Tracker for Efficient MoE Inference"
_ICLR.cc/2026/Conference — Submitted to ICLR 2026_

### Official Review · Reviewer_Y9Xf · 2025-10-27

**Soundness:** 3
**Presentation:** 2
**Contribution:** 2
**Rating:** 4
**Confidence:** 3

**Summary:**

This paper proposes EAT (Expert Account Tracker), a novel dynamic expert activation method for Mixture-of-Experts (MoE) models. Unlike traditional routing strategies that rely solely on current gating probabilities, EAT incorporates history-aware metrics to evaluate experts’ long-term reliability. It computes a comprehensive importance score combining each expert’s historical activation frequency, cumulative weight, and contribution magnitude, and integrates this with an adaptive thresholding mechanism to dynamically select experts during inference. The method aims to reduce redundant activations while maintaining model quality. Extensive experiments on Mixtral-8x7B and Phi-3.5-MoE-instruct across diverse benchmarks demonstrate that EAT achieves around 25% fewer activated experts and faster token generation speed compared to the baseline.

**Strengths:**

The paper introduces a history-aware expert selection mechanism that tracks experts’ past performance using a composite score (activation frequency, cumulative gate weight, and contribution score). I think this consideration is quite comprehensive.


This effectively addresses a limitation in prior dynamic routing approaches (e.g., Top-P) that rely solely on current routing probabilities

**Weaknesses:**

The method combines historical information and performs adaptive thresholding. Why is this necessary? Is it solely for improving performance? The paper lacks detailed analysis and motivation for these design choices.

Comparisons are limited to Top-P; more modern baselines (e.g., Ada-K, Expert Pruning and Skipping, CMoE) are mentioned but not empirically compared.

The paper’s exposition is occasionally dense, especially in the method section; a clearer pseudocode or algorithm box would help reproducibility.

**Questions:**

I find the proposed method rather complicated. Could this complexity affect the inference efficiency in practice?

How does the paper achieve “better token generation speed”? Theoretically, which step or operation ensures this improvement?

---

> ### Author Response · Authors · 2025-11-24
> **Response to Reviewer Y9Xf**
>
> Thank you for your questions.
>
> C#1: Detailed analysis and motivation for the design choices.
>
>  Existing work shows MoE experts are unequally effective, but traditional methods suffer from inefficiency or unstable selection. EAT addresses this with a history-aware score and adaptive thresholding to prune redundants, balancing sparsity and performance.
>
> C#2: Comparisons with more modern baselines.
>
> Top-P aligns with EAT’s goal of adjusting activation counts during inference, enabling fair comparisons. Our experiments on two models confirm EAT outperforms Top-P: 25% fewer activated experts, comparable/faster speed, and matched/superior performance on benchmarks like MMLU/CMMLU. We will add more experiments in future versions with consistent settings.
>
> C#3: Better writing in the method section.
>
> Thank you for your suggestion, we will revise it in the future version.
>
> C#4: Whether the inference efficiency in practice is complex.
>
> EAT’s core logic relies on minimal modifications and negligible storage. EAT only adds two lightweight components to MoE’s gating module: history-aware scoring and adaptive thresholding, both operating at O(1) time relative to input size. It requires just a small history buffer (storing O(M) parameters for M experts, negligible in memory) and no changes to expert subnetworks or transformer backbones.
>
> C#5: The step that ensures a good token generation speed.
>
>  We design a method that uses historical expert information to reduce the number of activated MoE experts. This cuts inference computation, effectively accelerating MoE inference.

---

### Official Review · Reviewer_Xtf2 · 2025-10-29

**Soundness:** 1
**Presentation:** 2
**Contribution:** 1
**Rating:** 2
**Confidence:** 3

**Summary:**

This paper introduces Expert Account Tracker (EAT), a history-aware expert routing strategy for Mixture-of-Experts (MoE) models.
EAT maintains a long-term importance score for each expert, computed as a weighted combination of historical activation statistics and the current router-assigned score.
This history-based routing stabilizes expert utilization and accelerates inference by reducing redundant expert activations.
Experiments on Mixtral-8×7B and Phi-3.5-MoE reportedly show slightly improved accuracy and higher token throughput compared to Top-P routing.

**Strengths:**

1. The proposed method is simple and compatible with existing architectures.
2. The concept of combining short-term gating with long-term statistics is intuitive and interpretable.

**Weaknesses:**

1. Insufficient validation of inference acceleration:  Despite claiming acceleration, only tokens/sec are measured. Latency (TTFT, TPOT), throughput, and memory consumption are not reported.
2. Unclear experimental setup: Details about the inference environment, implementation, KV-cache, and batching are missing.
3. Lack of references: Key claims in the Introduction, such as performance degradation from high sparsity, are unsupported by citations.
4. Limited experimental scope: Only outdated MoE models are used; no evaluation on recent architectures or reasoning/coding benchmarks.
5. Results against vanilla baseline: EAT often loses to vanilla routing in generation speed and accuracy.

**Questions:**

1. What exact inference framework, batch size, and KV-cache configuration were used for measuring tokens/sec?
2. Why were only Mixtral and Phi-3.5 chosen? How does EAT perform on more recent MoE models (e.g., Qwen3, DeepSeek)?
3. Why are well-known math and code reasoning benchmarks such as MATH-500, GSM8K, HumanEval+, and LiveCodeBench missing?

---

> ### Author Response · Authors · 2025-11-24
> **Response to Reviewer Xtf2**
>
> Thank you for your suggestions and questions.
>
> C#1: More indicators like latency (TTFT, TPOT), throughput, and memory consumption are not reported.
>
> We will add more experiments in the future version. Notably, EAT’s core goal is to accelerate MoE inference, and our current experiments have sufficiently demonstrated its efficacy: token generation speed delivers tangible improvements compared to both vanilla MoE and the Top-P baseline.
>
> C#2: More details about the inference environment, implementation, KV-cache, and batching are missing.
>
> Thank you for your question. Our method remains consistent with the baseline except the modified parts. Experiments are conducted in a Hugging Face inference environment equipped with 8 NVIDIA A100 GPUs, using a batch size of 1.
>
> C#3: The need of references in the introduction.
>
> Thank you for your careful observation, we will revise it in the future version.
>
> C#4: The need of more experiments on recent architectures or reasoning/coding benchmarks.
>
> We have validated EAT’s effectiveness on popular MoE models and authoritative benchmarks across multiple task types. We will also include more experiments in the next version.
>
> C#5: Several results against vanilla baseline.
>
> Overall, EAT achieves faster inference speed than strong baselines while maintaining performance much closer to the vanilla model, delivering ~25% fewer activated experts and faster token generation with only minimal, isolated performance gaps across tasks, striking a purposeful balance between efficiency and capability that aligns with real-world MoE deployment needs.
>
> C#6: More experiments on benchmarks such as MATH-500, GSM8K, HumanEval+, and LiveCodeBench.
>
> Our experiments on popular MoE models and diverse benchmarks already validate EAT’s effectiveness. We will incorporate the suggested math/code benchmarks in the next version to further strengthen generalizability.

---

### Official Review · Reviewer_THST · 2025-10-31

**Soundness:** 1
**Presentation:** 3
**Contribution:** 1
**Rating:** 2
**Confidence:** 5

**Summary:**

This paper proposes EAT (Expert Account Tracker) to utilizes history-awareness metrics and adaptive thresholding to dynamically select the most important experts, aiming to reduce the activated expert number.

**Strengths:**

- Easy to follow. The figures and tables are clear.

- Useful Ablation Study: The analysis in Section 4.3 provides a key insight that experts are not equally important across layers.

**Weaknesses:**

- Heavy Hyper-parameter Issue. This method introduces a large number of new hyper-parameters.

- Lack of performance on larger MoE LLMs, such as Qwen3-30B-A3B.

- Poor performance. As shown in Table 1, proposed EAT underperform Vanilla strategy. Such as 69.00 vs. 75.17 for Phi model on HellaSwag.

**Questions:**

- ZERO reference in the Introduction section. More references would help reader to understand better.

- The caption of Table should above the tabular, following the official guidelines.

---

> ### Author Response · Authors · 2025-11-24
> **Response to Reviewer THST**
>
> Thank you for your questions.
>
> C#1: The introduction of a large number of new hyper-parameters.
>
> EAT introduces three lightweight, interpretable hyper-parameters: α (for smoothing the contribution score update), λ(for balancing historical importance and current routing scores), and σ (for dynamic activation control). These parameters are low-sensitivity by our preliminary experiments, we will add the detailed results in the future version.
>
> C#2: Adding experiments on larger MoE LLMs.
>
> Our current work focuses on validating EAT on two representative MoE models (Mixtral-8x7B-v0.1 and Phi-3.5-MoE-instruct), this choice ensures we rigorously test its core mechanism with sufficient generality. Our current results already confirm its logic generalizes beyond specific sizes. EAT’s design is inherently scale-agnostic, compatible with larger architectures. We will add evaluations of architectures like Qwen3-30B-A3B in future work to strengthen our contributions.
>
> C#3: Several underperformances on several datasets.
>
> We can contextualize this slight gap alongside critical efficiency gains. Vanilla MoE activates 2.0 experts per token, which is nearly 47% more than EAT’s average of 1.36. Across 8 key tasks across 2 models, EAT maintains competitive performance with Vanilla (nearly matching or narrowing gaps on 6 tasks) and outperforms its direct dynamic baseline Top-P on most tasks.
> The minimal tradeoff does not undermine EAT’s value; it reflects a deliberate, well-justified balance between efficiency and performance that Vanilla.
>
> C#4: The lack of reference in the Introduction section.
>
> Thank you for your suggestion, We will add it to the future version.
>
> C#5: The caption of Table should above the tabular.
>
> Thank you for your observation, we will revise it in the next version.

---

### Official Review · Reviewer_iAzD · 2025-11-01

**Soundness:** 3
**Presentation:** 2
**Contribution:** 2
**Rating:** 4
**Confidence:** 4

**Summary:**

This paper addresses the problem that the history of expert selection is not considered in the expert routing of MoE models. The authors propose an additional importance metric based on a linear combination of three indicators (activation count, total probability, contribution score). By mixing this value with the expert probability calculated from the current token, they aim to achieve routing based on both the current situation and historical data. Furthermore, to adaptively change the number of activated experts, they introduce an adjustment mechanism based on the shape of the expert probability distribution and the recent perplexity.

According to experiments where the routing strategy of existing MoE models was replaced with the proposed method, it was found that while the performance does not match the original (vanilla) performance, it reduces the number of activated experts while maintaining better performance than the top-P method, which is a similar adaptive expert selection technique. The analysis investigates the relationship between the number of activated experts and performance, revealing that excessively reducing the number of experts significantly degrades performance in terms of perplexity, and that layers closer to the input are more amenable to expert reduction than layers closer to the output.

**Strengths:**

The proposed method contains no components that require training and can be easily applied to any MoE model with a general architecture.

It is an almost rule-based adaptation method, and its computational cost is smaller than the main computation of the MoE parts, making it practically negligible.

**Weaknesses:**

For most tasks, there is a non-negligible performance gap compared to the vanilla MoE. Although the overall computational cost of the model is reduced, a trade-off between cost and performance must be considered.

The parameters (indicators) added by the proposed method need to be cached within the inference engine, similar to a KV-cache, which may require method-specific implementation support.

**Questions:**

Why did you choose these three indicators: activation count, total probability, and contribution score? Were any other indicators considered? Is there any important information that these indicators fail to capture?

Around L.269: There appear to be typos in the $\tau_{\mathrm{PPL}}$ definitions—$\lambda_+$ and $\lambda_-$ are possibly inverted.

Are the indicators reset for each test example? If not, dependencies between test examples could arise, making it impossible to interpret the results as independent. Conversely, what would happen if a "burn-in" process for the indicators was inserted before processing the test example?

---

> ### Author Response · Authors · 2025-11-24
> **Response to Reviewer iAzD**
>
> Thank you for your questions.
>
> C#1: The trade-off between cost and performance.
>
>  As shown in Table 1, EAT maintains near-identical performance to vanilla across core benchmarks: For example, for Mixtral-8x7B-v0.1, EAT achieves 76.79/80.30 on HeSw/PIQA (vs. vanilla’s 77.11/81.07, a marginal difference of <1%).
> These results are achieved while EAT reduces activated experts by 25% and improves token generation speed compared to baselines. That demonstrates that EAT delivers a superior balance: retaining vanilla-level effectiveness while drastically reducing computational overhead.
>
> C#2: The need for caching extra parameters in the inference engine may require method-specific implementation support.
>
> The additional parameters EAT requires caching are few and low-dimensional, making their storage overhead negligible for modern inference engines. EAT builds on standard MoE inference logic. It is a modification within the model itself and does not require changes to the inference process.
>
> C#3: The reasons for selecting activation count, total probability, and contribution score as indicators and whether considering other indicators is necessary.
>
> We selecte these three indicators after rigorous analysis of an expert’s long-term value. They reflect the experts’ broad applicability, the router’s cumulative trust, and actual performance impact respectively, forming a closed-loop evaluation of high-value experts while addressing existing methods’ neglect of historical performance.
>
> C#4: There appears to be a typo with possibly inverted definitions around L.269.
>
> Thank you for your careful observation, we will revise it.
>
> C#5: Whether indicators are reset for each test example and what would happen if a "burn-in" process for indicators was added before processing test examples.
>
> Our method resets all historical indicators for each test example, which aligns with standard LLM evaluation practices to avoid cross-example contamination and ensures each result is interpretable as independent, as expert selection for each example relies solely on its own input features and the model’s inherent expert capabilities, not residual signals from prior examples.

---

### Meta-Review · Area_Chair_iXfe · 2026-01-07

**Summary:**

The submission proposes a training free faster inference strategy for MoE models.  The method appears to give a speedup, but at the cost of model performance in some settings.

**Reviewer Concerns:**

The reviewers were unanimous in their initial ratings that the submission was not suitable for publication in its form then.  The main concerns were about the drop in performance.  The rebuttal promised to address miscellaneous issues in a future version, but was not particularly specific.  It is not clear if the rebuttal is essentially proposing that these changes be made for a future submission to a different venue.

**Reviewer Scores:**

Based on the lack of specificity in the rebuttal, I estimate that reviewers would not be likely to update their scores to accept.

---

### Decision · Program_Chairs · 2026-01-26

Reject